# A Modified FlowDroid Based on Chi-Square Test of Permissions

**DOI:** 10.3390/e23020174

**Published:** 2021-01-30

**Authors:** Hongzhaoning Kang, Gang Liu, Zhengping Wu, Yumin Tian, Lizhi Zhang

**Affiliations:** School of Computer Science and Technology, Xidian University, Xi’an 710071, China; kanghzn@stu.xidian.edu.cn (H.K.); wuzhenping@stu.xidian.edu.cn (Z.W.); ymtian@mail.xidian.edu.cn (Y.T.); rnzhang@stu.xidian.edu.cn (L.Z.)

**Keywords:** automatic control, mutual information, static detection, Chi-square test, permission, FlowDroid

## Abstract

Android devices are currently widely used in many fields, such as automatic control, embedded systems, the Internet of Things and so on. At the same time, Android applications (apps) always use multiple permissions, and permissions can be abused by malicious apps that disclose users’ privacy or breach the secure storage of information. FlowDroid has been extensively studied as a novel and highly precise static taint analysis for Android applications. Aiming at the problem of complex detection and false alarms in FlowDroid, an improved static detection method based on feature permission and risk rating is proposed. Firstly, the Chi-square test is used to extract correlated permissions related to malicious apps, and mutual information is used to cluster the permissions to generate feature permission clusters. Secondly, risk calculation method based on permissions and combinations of permissions are proposed to identify dangerous data flows. Experiments show that this method can significantly improve detection efficiency while maintaining the accuracy of dangerous data flow detection.

## 1. Introduction

Google Android is a mobile operating system that is widely used in many fields [1,2]. With the development of the Internet of Things, Android quickly gained a large proportion of the market share. At the same time, the number of malicious applications (apps) has been increasing [3] and over the last few years, the amount of malware has increased significantly. According to a recent report from McAfee, over 1.6 million new examples of mobile malware were discovered in the first quarter of 2019 [4]. Therefore, the detection of Android malware with a high accuracy rate and high efficiency is an important issue.

Various approaches have been proposed in previous works with the intention of detecting Android malware. These approaches can be categorized into static analysis, dynamic analysis or hybrid analysis [5]. Dynamic analysis means that, in the process of running an application, the flow of privacy information and data is tracked and captured, and the malicious tendency of application behavior is analyzed and judged. Dynamic analysis can monitor and track the flow of private data in real time [6] and is not affected by code obfuscation, encryption and other factors. However, privacy leaks that are not triggered at runtime cannot be detected, and the low code coverage causes a high missing rate. At the same time, real-time operation results in greater resource consumption [7]. In the case of resource shortages on mobile devices, the system efficiency will be seriously affected. In contrast to dynamic analysis, static analysis is done without running an app. In static analysis, features such as permissions and API calls are extracted from the app source code by reverse engineering to analyze and infer suspicious behavior from an app and discover problems in different stages of the entire life cycle, verifying the security of app at the source code level. As a highly influential static analysis tool for Android apps, FlowDroid [8] has the advantage of wide code coverage and it can detect many malicious behaviors that cannot be detected by dynamic analysis. However, source code level analysis will bring a large amount of irrelevant detection while leading to high false positives and low detection efficiency, decreasing the availability of these tools. In our experiments, for apps over 10 MB in size, FlowDroid reports timeouts and insufficient memory. Even for the two apps from the FlowDroid samples, the test takes nearly an hour.

Research and experiments show that app security threats are strongly correlated with some characteristic permissions [9]. When using FlowDroid, only those flow paths that actually cause privacy leakages need to be considered, which significantly reduces the scale of analysis and improves efficiency. This paper proposes a redundancy resolution method for FlowDroid, which can cluster the correlated permissions and calculate the risks of flow paths. This paper provides the following two contributions:(1)We propose a permission clustering method based on the use of permissions by malware. Compared with all the permissions of the Android system, the proposed permission cluster contains only a few permissions. By monitoring these permissions and the related call paths in FlowDroid, the analysis can be greatly simplified.(2)We propose a lightweight malicious application detection method based on the permission clusters and FlowDroid in this article. This method is suitable for pre-installed software and user-installed software. We improved FlowDroid so that it can only monitor the call paths that are related to the permissions in the permission cluster and meet certain risk conditions. This greatly reduces the detection time and memory usage.

## 2. Related Work

The static analysis of Android malware relies on Java bytecode, which is extracted by disassembling an app. The manifest file is also a source of information for static analysis. Kirin [10] is the safety inspection scheme for app installation, which operates by defining security rules to identify dangerous permission combinations; the installation strategy is formulated based on the use of security rules as detection criteria. However, due to the small number of rules and the lack of representation of permission combinations, the detection efficiency and the accuracy cannot be guaranteed. TrustDroid [11] provides two alternative detection modes: real-time detection on the mobile device side and static analysis on the server side, converting the data flow to a tree structure using Jasm in middle code representation to generate a function call graph, preventing untrusted apps from leaking user privacy information. The resource consumption of TrustDroid is also extremely high. LeakMinder [12] analyzes the security of apps from the third-party market and decompiles Android application package (APK) files by reverse engineering. Based on a predefined source and sink, LeakMinder generates a call graph and data flow diagram and finds possible privacy leak paths. However, implicit data leaks cannot be detected. Besides, designed artificially sources and sinks are not particularly representative, which results in contingency and inaccuracy. Cen et al. [13] proposed the use of a probabilistic discriminative model based on regularized logistic regression for Android malware detection. The probabilistic discriminative model works well with permissions and achieves the best detection results by combining both decompiled source code and application permissions. Kang et al. [14] proposed a method that detects and classifies Android malware using static analysis with the combination of the attacker’s information. The effectiveness of Android malware detection is improved by integrating the attacker’s information as a feature, and the method categorizes illegitimate applications into homogeneous classes. Song et al. [15] proposed an integrated static framework using a filtering technique consisting of four layers to identify and evaluate mobile malware on Android. Sun et al. [16] presented an approach that interfaces static logic-structures and dynamic runtime information to detect Android malware. Behavior similarity is used for the classification of malware. The results showed that the approach is easy to implement and has low computational overheads. Rovelliet al. [17] presented a permission-based malware detection system that uses machine learning classifiers on the behavioral patterns to consequently distinguish inconspicuous applications. DAPASA [18] is an approach used to detect Android piggy-backed apps through sensitive subgraph analysis. DAPASA generates a sensitive subgraph (SSG) to profile the most suspicious behavior of an app. Five features are constructed from SSG to depict the invocation patterns. The five features are fed into the machine learning algorithms to detect whether the app is piggy-backed or benign. Talha et al. [19] presented a permission-based Android malware detection system consisting of three components, namely the central server, Android client and signature database, and static analysis is used to categorize the Android application as normal or harmful. Li et al. [20] raised the issue of considering interaction terms across features for the discovery of malicious behavior patterns in Android applications and proposed a classier for Android malware detection based on a factorization machine architecture.

FlowDroid [8] was proposed by Arztet al. in 2013 and has been widely studied and applied in the field of Android static analysis. FlowDroid is considered as a context, flow, field and object-sensitive and lifecycle-aware static taint analysis tool for Android apps. To increase recall, FlowDroid creates a complete model of Android’s app lifecycle. However, a large number of normal paths are also detected while the entire life cycle is analyzed, causing false positives and low efficiency. The main purpose of this paper is to improve the analysis efficiency and applicability of FlowDroid.

## 3. Preliminaries

### 3.1. Android Permission

The Android system is an extension based on Linux, which provided the permission mechanism [21]. Operations that apps can perform are specified to limit the software’s ability to manipulate systems or other software. The Android permission mechanism requires developers to apply for permissions they need in Android’s Manifest.xml and gets user’s consent during installation to access system resources and functional components by calling the related API. Android protects sensitive systems and user information by restricting apps from accessing system resources with permissions other than those declared. Android uses a coarse-grained permission management mechanism and no longer reviews the running process after granting permissions; thus, malicious apps exploit users’ ignorance of permissions and the coarse-grained permission management of Android’s permission mechanism to access or even leak sensitive information. Android 8.0 provides 135 permissions and corresponding APIs to access system resources. In fact, only a small portion of permission usage can lead to sensitive information leakage. If malicious application-independent permission calls are accurately excluded from detection, the data paths that need to be detected in static analysis can be significantly reduced, thus reducing the false alarm and improving detection efficiency.

### 3.2. FlowDroid

FlowDroid, based on Soot [22], works directly at the bytecode level and does not require access to an app’s source code. It parses the APK file of an Android app, converts the Java code into Jimple middle code, simulates the life cycle of an Android app to handle callback functions and generates a call graph (CG) and inter-procedural control-flow graph (ICFG) [23,24] to trace taints (Figure 1). It uses the Interpretural Finite Distributive Subset (IFDS) to model data flow propagation and generates complete, polluted data flow paths by the Heros framework [25]. Therefore, FlowDroid has very high requirements in terms of computing and memory resources. For Enriched1.apk (a sample application of which is shown in [25]), a total of 46 nodes and 78 function call paths were generated (Figure 2 shows a partial call graph of these). The call graph composed of the data paths to be analyzed is very complex, and there is a large number of callbacks and callback relationships between functions, which leads to high time and resource costs.

Our experiments show that FlowDroid usually reports timeout or out-of-memory errors for apps with a size larger than 10 MBytes. The reason for this is that an Android app often involves dozens of components at runtime, and the interaction between multiple components leads to hundreds of callback methods.

Although full-scale analysis can ensure high accuracy, it results in an unnecessary amount of analysis. FlowDroid should be improved in two aspects as follows:(1)There is no further analysis of taint paths, and the large number of false positive paths results in low accuracy.(2)There is no clear analysis content and there are no taint path identification criteria. This leads to a large number of irrelevant detection paths, resulting in excessive memory and time consumption.

### 3.3. Mathematical Background

The Chi-square test is a hypothesis testing method used to determine whether two variables are independent. For two discrete variables, it can be concluded whether there is a correlation between them by using the Chi-square test. The larger the Chi-square value, the greater the deviation between the two variables, the smaller the correlation between them and the stronger the independence. When the value of Chi-square reaches 0, that means the factors are exactly the same. The formula of the quaternary Chi-square test is as follows:(1)$$\chi^{2}=\frac{N{\left(AD-BC\right)}^{2}}{\left(A+B\right)\left(C+D\right)\left(A+C\right)\left(B+D\right)}$$

For an abstract random variable, to remove its uncertainty, a certain amount of information needs to be used, and information entropy is a mathematical measure of this. The higher the information entropy is, the larger the amount of information that needs to be introduced and the lower the information entropy is, and the less information is needed. The information entropy of *X* is defined as:(2)$$H\left(X\right)=-\underset{n}{\sum }P\left(X\right){log}_{2}P\left(X\right)$$

In order to determine the influence of the information entropy between two variables, the information entropy of *X* can be obtained when *Y* appears, as shown in Equations (3) and (4):(3)$$H\left(X|Y\right)=-\underset{n}{\sum }P\left(Y\right)\underset{n}{\sum }P(X|Y){log}_{2}P(X|Y)$$
(4)H(X,Y)=H(X)+H(Y|X)=H(Y)+H(X|Y)
where H(X|Y) is the conditional information entropy, P(X|Y) is the conditional probability and H(X,Y) is the joint information entropy. According to the above equations, the mutual information values of *X* and *Y* can be obtained as I(X;Y):(5)$$I(X;Y)=H(X)-H(X│Y)=-\sum^{\;}P\left(X,Y\right)log\frac{P\left(X,Y\right)}{P\left(X\right)P\left(Y\right)}$$

## 4. The Improved Detection Method

Flowdroid analyzes all data paths, resulting in high false positives and high resource requirements. This paper presents a redundancy resolution method based on feature permissions and risk. The purpose is to exclude the large number of irrelevant paths (security paths) for static analysis. The lightweight FlowDroid, named Permission-based FlowDroid (PBFlowDroid), is proposed based on above methods. Figure 3 shows the architecture of PBFlowDroid.

Firstly, the Chi-square test is used to extract permissions related to malicious applications, and these permissions (malicious sensitive permissions) are classified into permission clusters by a clustering algorithm based on mutual information. Thus, the large number of permissions is reduced to a small number of permission clusters which are considered in static analysis. Secondly, different permissions or combinations of permissions bring different risks to user privacy or system security. To improve the accuracy of analysis, a risk assignment and calculation algorithm for single permissions or combinations of permissions is proposed. With these methods, all paths are given a risk value. Using the risk value of each path, the security of the taint data flow propagation path generated by control flow [26] and data flow can be determined, notifying the user whether the taint data flow is a safe path, ensuring the accuracy of static analysis and improving the analysis efficiency.

### 4.1. Permission Cluster Extraction

In Android, each permission has the two states of “request” and “no request”, which are independent of the number of requests. This scenario is suitable for the Chi-square test. In this study, the quaternary Chi-square test is used; the Chi-square of permission *p* is as follows:(6)$$\chi^{2}\left(p\right)=\frac{N{\left(A_{p}D_{p}-B_{p}C_{p}\right)}^{2}}{\left(A_{p}+B_{p}\right)\left(C_{p}+D_{p}\right)\left(A_{p}+C_{p}\right)\left(B_{p}+D_{p}\right)}$$
where *N* denotes the total number of app samples, which consists of *X* malicious apps and *Y* normal apps. For permission *p*, requests by malicious apps and normal apps are counted as $A_{p}$ and $B_{p}$. The numbers that do not apply for *p* by malicious apps and normal apps are counted as $C_{p}=\left(X-A_{p}\right)$ and $D_{p}=\left(Y-B_{p}\right)$, as Table 1 shows.

As for $\chi^{2}$,the Chi-square test provides a threshold checklist as a criterion of reliability. For each permission, the probability of it relating to a malicious application is obtained by referring to the Chi-square test threshold table [27]. The larger the probability, the more malicious applications tend to have the corresponding permission, while a Chi-square value less than 0.5 indicates that the permission has almost no correlation with malicious applications. In our experiment, the top 20 permissions with a Chi-square value greater than 0.5 are regarded as permissions with a high correlation with malicious applications, as shown in Table 2.

The Chi-square test selects the permissions to be investigated and significantly reduces the candidate paths for static analysis. Even with 20 permissions, the number of paths that can be associated with some apps is still quite large. In fact, permissions are not independent of each other. When one app applies for a certain permission, other permissions of the same type that are related to achieve a combined function are also applied, which leads to strong correlation between permissions. For example, “READ_SMS” and “WRITE_SMS” are often applied and used at the same time. In static analysis, if two permissions with high correlation are detected separately, multiple detection results will be generated. This may decrease the accuracy of detection. To solve this problem, we use the clustering algorithm to cluster the selected permissions so that each cluster is representative.

Permission is a discrete kind of feature information, and the similarity between permissions can be measured by mutual information based on information entropy. We use $P_{m}\left(X\right)$ and $P_{n}\left(X\right)$ to represent the probability that permission *X* will be maliciously applied and normally applied, respectively. Then, the information entropy *H (X)* of permission *X* is as follows:(7)$$H\left(X\right)=-\left(P_{m}\left(X\right){log}_{2}P_{m}\left(X\right)+\;P_{n}\left(X\right){log}_{2}P_{n}\left(X\right)\right)$$

The mutual information values of permissions *X* and *Y* are calculated by formula (5). In order to describe the similarity between permissions *X* and *Y* more intuitively, the correlation between the two permissions can be obtained by (8):(8)$$Cor\left(X,Y\right)=2\times \left[\frac{I\left(X,Y\right)}{H\left(X\right)+H\left(Y\right)}\right]$$
where the value of Cor(X,Y) is located between [0, 1]. A value of 0 means that permission *X* and *Y* are completely unrelated; the larger the value, the greater the correlation between them.

In this paper, a clustering method based on mutual information is proposed to cluster the selected permissions (which in our experiment, the number of permissions is 20 as shown in Table 2) to generate feature permission clusters (FPC) with low similarity between clusters and high similarity within clusters and further remove irrelevant detections in static analysis.

The steps of the clustering algorithm based on mutual information are as follows:


**Clustering Algorithm Based on Mutual Information:**

*Input: Permissions set Obtained by Chi-Square Test:*
$S=\left\{p_{0},p_{1},\cdots ,p_{n}\right\}$

*Output: Cluster Sets:*
$C=\left\{c_{0},c_{1},\cdots ,c_{m}\right\},\;m\leq n$

(1)Take the first element $p_{0}$ in S as the first element of $c_{0}$, and delete $p_{0}$ from S.(2)For each
$p_{i}$ in S(3)for each $c_{j}\;\neq \;null$ in C do(4)if
$Cor(c_{j},\;p_{0},\;p_{i})\;>\;T$ then(5)add $p_{i}$ to $c_{j}$;(6)else(7)create a new cin C, and add $p_{i}$ to c;(8)end if(9)end for(10)end for(11)for each $c_{i}$ in C(12)if $Num(c_{i})\;<\;$3 then(13)cluster the similar Cor in $(c_{i}$, and put it to a new *c* in *C*(14)end if(15)end for(16)output *C*


After clustering, the permissions with high correlation with malicious apps were clustered into multiple clusters (we have seven clusters in our experiment, from *c*_0_~*c*_6_, as shown in Table 2).

FPC extraction not only eliminates irrelevant paths caused by permissions unrelated to malicious applications but also eliminates the correlation within the permission cluster, further eliminating the redundancy in static detection and improving the detection efficiency.

### 4.2. Risk Calculation

App security threats are strongly correlated with some characteristic permissions [9]. Operations corresponding to different permissions pose different threats to user privacy and system security [9]. For example, the operation of applying for network permission often transfers privacy settings on a device to other addresses involving the interaction between user information and the outside world, and so the threat degree is greater; the permission of applying for the device’s local location only obtains the current user’s status and does not interact with the outside world or affect the security of the device, so the threat degree is general.

To visualize the risk level of different permission clusters, we use the risk value to describe it. The risk value is the quantification of the risk of each permission cluster, and the risk value of each permission in the same cluster is the same. The selection of these values is not unique, and only three conditions need to be met:(1)The risk value decreases as $\chi^{2}$ of permission cluster decreases;(2)The risk values of $c_{0}$ to $c_{k-1}$ are greater than 1;(3)The risk values of $c_{k}$ to $c_{m}$ are greater than 0;

Where *m* + 1 is the number of cluster sets, and *k* is used to classify clusters into high risk clusters and low risk clusters. In our experiment, the *m* is 6 and we choose 4 as the value of *k*.

These limitations are related to analytical calculation methods. When assigning risk values to each cluster, in this paper, we select a simple assignment that satisfies the above three conditions. It should be noted that the threshold value in 5.1 is related to the risk value; this is an empirical value obtained through experiments. Different risk value assignments will correspond to different threshold values. In our experiment, the risk value for each cluster is shown in Table 2.

When an app performs an operation or acquires a resource, it sometimes applies for more than one permission, which forms a combination of permissions. A combination of permissions may pose a greater threat to the system than a single permission [28]. In our experiment, we refer to [10] to calculate the risk value of an app, and the calculation rules are as follows:

*Rule 1*: For a single requested permission, the risk value $R_{S}$ is the sum of the risk value for each permission:(9)$$R_{s}=\overset{M}{\underset{i=1}{\sum }}R\left(p_{i}\right)$$
where $R\left(p_{i}\right)$ is risk value of permission $p_{i}$, and $p_{i}$ is the single permission requested by the app.

*Rule 2*: For any requested combination of permissions $PC_{j}$ with where permissions belong to clusters $c_{0}$ to $c_{k-1}$, the risk value $R_{C}\left(PC_{j}\right)$ is defined as:(10)$$R_{c}\left(PC_{j}\right)=\{\begin{array}{c} \prod^{\;}R\left(p_{i}\right),\;p_{i}\in c_{j}\;and\;j<k\\ \sum^{\;}R\left(p_{i}\right),\;p_{i}\in c_{j}\;and\;j\geq k \end{array}$$
and the risk of the combined permissions is defined as the sum of all risk values of each combination of permissions; that is,
(11)$$R_{c}=\;\overset{m}{\underset{j=0}{\sum }}R_{c}\left(PC_{j}\right)$$

*Rule 3*: The risk value *R* of an app is defined as the logarithmic mean of the total risk value:(12)$$R=\frac{log\left(R_{S}+R_{C}\right)}{M}$$
where *M* is the total number of requested single permissions and combinations of permissions.

In general, a normal app provides multiple services to satisfy users’ functional needs, and several permissions are used, whereas a malicious app has simpler functions but uses permissions with a higher risk value [29]. Therefore, we use the mean risk rather than the total risk to evaluate the risk of an app. For apps whose *R* is less than a threshold, it can be preliminarily judged as a secure application. For apps with a higher *R* than the threshold, the mapping between the permission and corresponding API is constructed and added to the *Source* set, then the security decision of the taint data flow is entered.

### 4.3. Filtration of Taint Data Flow

The function call graph generated by FlowDroid is very complex. Frequent calls between functions cause a large number of redundant detection paths, which makes the further static analysis cost very high. In this study, data paths from native FlowDroid are further filtered based on the risk value.

On the basis of the definition of native FlowDroid, we define the following extra variables:(1)*Route*: the taint transmission path set. The element is a pair—<*source*, *sink*>—and it indicates that there is a taint transmission path from *source* to *sink*, where source∈Source and sink∈Sink.(2)*LeakRoute*: the privacy leak path set. The element is a triple—<*source*, *sink*, *risk*>—and it indicates that there is a leak path from *source* to *sink* with a risk value of *risk*. The filtering method is shown in Figure 4.

For the filtering method in PBFlowDroid, the criterion is whether the risk value of the path is higher than the threshold. Based on the risk calculation for the path in the call graph, the filtering method extracts the paths with a higher risk value to *LeakRoute*. In this way, we obtain all the pollution paths and identify the data paths with a high risk of privacy leakage. Because the filtering method reduces the data paths to be analyzed, not only is the analysis time shortened, but also the accuracy of pollution analysis can be improved due to the elimination of false positive paths.

From the above, PBFlowDroid introduces a high-risk pollution path detection method and reduces the scale of pollution paths to be analyzed. Furthermore, PBFlowDroid can solve the challenge of the high false-positive and false-negative rate of analysis in native FlowDroid.

## 5. Experiments

This section tests the accuracy and efficiency of the proposed PBFlowDroid. The computer used was a Z600 WorkStation with an Intel (R) Xeon (R) E5540 @ 2.53 GHz CPU and 4.00 GB of physical memory. All tests were run on Windows 7 with Oracle’s Java Runtime version 1.8 (64 bit). Android 6.0 with Android-23 SDK was used in all experiments.

### 5.1. Accuracy Experiments

In this experiment, 500 normal apps from Google Play and 50 malicious apps from GitHub’s Malicious Application Sample Library [5] were used as test samples. The risk value *R* of each successfully tested app was calculated. An app with an *R* value less than the threshold was recognized as a normal application; otherwise, it was identified as a malicious application. The risk threshold was an unknown parameter at the beginning of the experiment. We were inspired by machine learning [30] and used one-tenth of the experimental data as a training set to obtain the risk threshold. The training set contained 50 randomly selected normal apps and five malicious apps. By calculating the risk value of all apps in the training set and selecting the threshold to minimize the false positive rate, we obtained the risk value threshold. Other data were used as the validation set. In our experiments, we set 0.12 as the threshold value. Table 3 shows the results.

In our experiments, 413 normal apps and 35 malicious apps were decompiled. Among the 413 normal apps, 332 were correctly identified and the other 81 were identified as malicious apps. The detection rate for the normal sample was 80.4%, and the false alarm rate was 19.6%. In total, 27 of 35 malicious apps were correctly identified. The omission ratio was 22.8% and accuracy rate was 77.2%. Compared with the test results of native FlowDroid in Droid Bench, where 30 malicious applications were detected out of 39 apps, with an accuracy of 76.9% [8], the proposed method guaranteed sufficient detection accuracy.

Table 4 gives the number of permissions in application as *M* and the taint paths and risk value as *R* for some apps. We can see that the *R* values of *FangTianxia* and *MeiPai* were 0.134 and 0.121, respectively; they were reported as malicious apps.

### 5.2. Efficiency Experiments

In PBFlowDroid, only data paths with a higher risk value are analyzed, which reduces the detection complexity. In this section, a comparative test with native FlowDroid in terms of detection time and memory consumption was performed. When we reproduced FlowDroid in our experimental environment, we found that FlowDroid took more than 10^4^ s to analyze some apps, and the magnitude of the results of the completed path analysis was generally between 10^4^ to 10^5^. Because the time complexity of FlowDroid is orders of magnitude different from the method proposed in this article, we do not compare the apps directly in this article. Thus, 500 test samples were randomly selected from Google Play with sizes ranging from 50 KB to 60 MB. For apps larger than 10 MB, FlowDroid reports a timeout or out-of-memory error. Table 5 shows the time and memory consumption of PBFlowDroid for eight apps.

With the increase of the app size, the memory consumption increases significantly for both tools. For FlowDroid, memory is quickly exhausted, making the detection fail. For PBFlowDroid, memory consumption is kept within 4 GB and all tests were successful in our experiments.

Table 6 lists the multiple classification algorithms supported by PUMA [31] and their results. However, machine learning algorithms, including the method proposed by PUMA, can only discriminate whether an app is a malicious app through the use of permissions and cannot analyze how apps abuse permissions. In contrast to machine learning algorithms, our method and FlowDroid can not only analyze whether an app is a malicious app but also analyze its usage of permissions. Moreover, the results in Table 5 show that our app analysis consumes less resources than FlowDroid.

## 6. Conclusions

FlowDroid is a static taint analysis tool widely used for Android apps. However, the call graph generated by FlowDroid grows exponentially as the size of the app increases, which reduces its availability. Research shows that the security threat of an app mainly comes from its abuse of permissions, and not all permissions will lead to a leakage of sensitive information. This paper proposes a method to identify dangerous data paths, and secure paths are filtered in further analysis. In this way, the call graph is greatly simplified and the resource requirements in the analysis process are significantly reduced. On the other hand, we used the Chi-square test and mutual information values to extract the correlated permissions and proposed risk calculation method considering permission combinations. In this way, a more accurate risk value is taken as a criterion to reduce misjudgment. The experimental results show that our proposed method reduces the complexity of detection significantly, and the detection accuracy is guaranteed.

In our future work, the communication between processes needs to be taken into account, and the assessment of communication risk is worth exploring. This will help to deal with the collusion attack problem. Second, the risk value of the data path should be determined based on API, application components and other features [32] rather than only permissions to improve the accuracy of pollution path identification. Third, the distinction between small malware and large malware should also be considered. We will import some large malware to the test set in our next work and prove the applicability of PBFlowDroid to large malware.

## Figures and Tables

**Figure 1 entropy-23-00174-f001:**
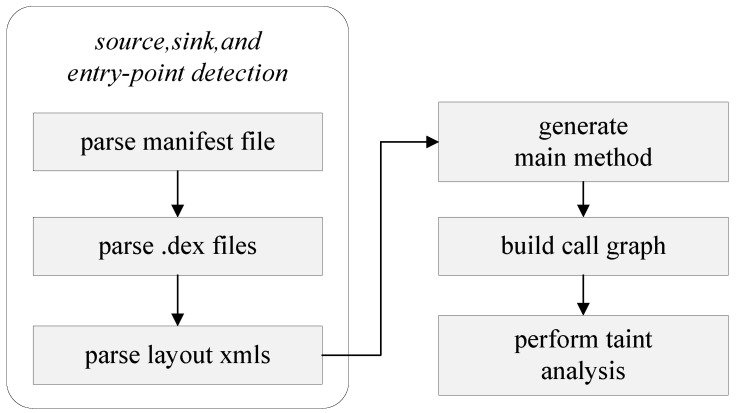
Workflow of FlowDroid.

**Figure 2 entropy-23-00174-f002:**
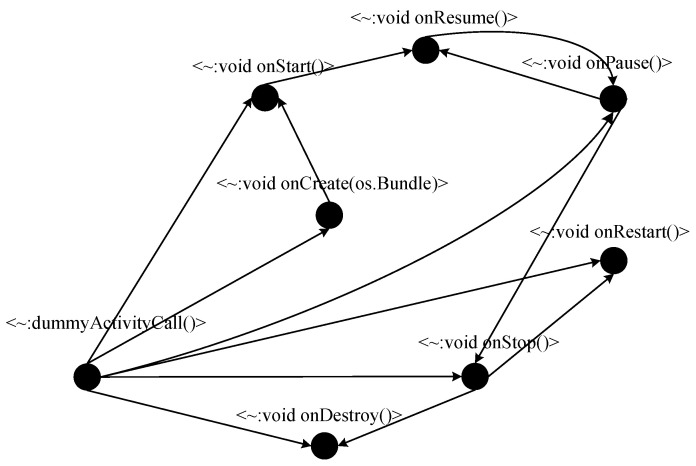
Partial function call graph of Enriched1.apk.

**Figure 3 entropy-23-00174-f003:**
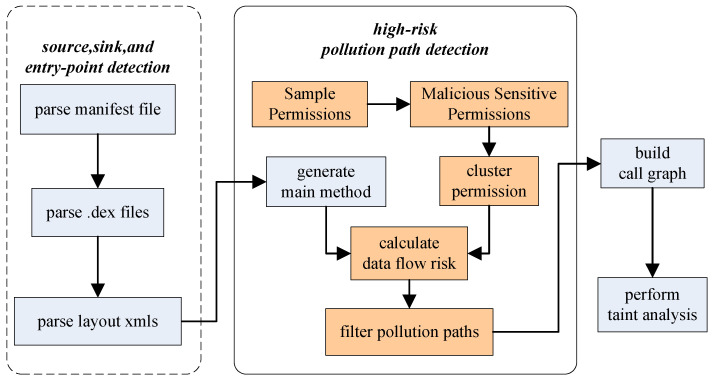
Workflow of PBFlowDroid.

**Figure 4 entropy-23-00174-f004:**
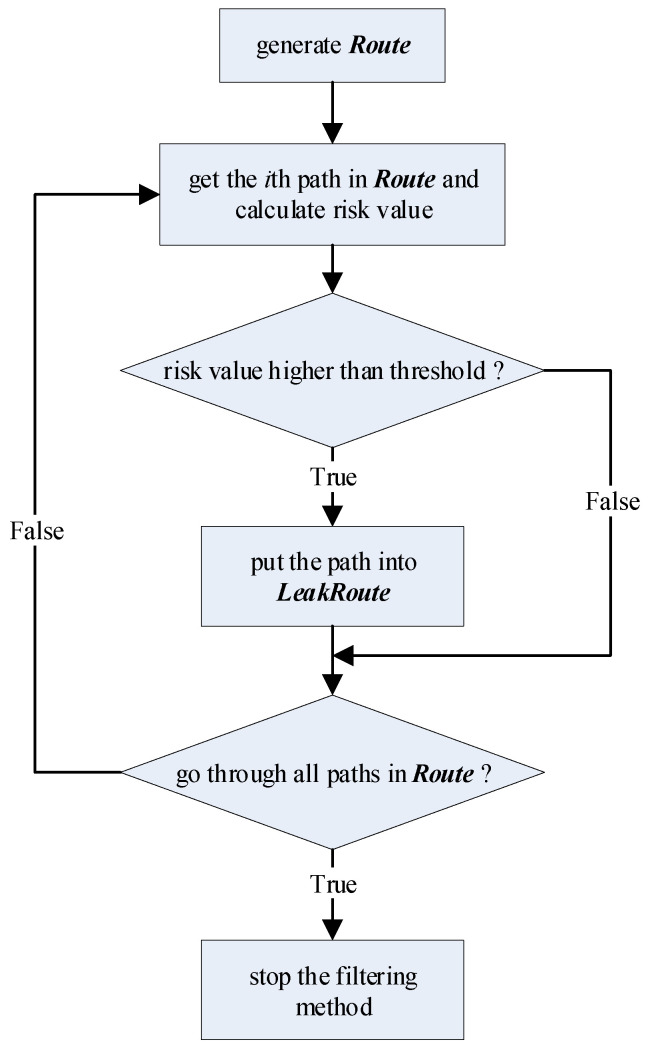
Workflow of filtering method.

**Table 1 entropy-23-00174-t001:** Chi-square test distribution of permission *p*.

	Number of Malicious Apps (*X*)	Number of Normal Apps (*Y*)
Apps with permission *p*	$A_{p}$	$B_{p}$
Apps without permission *p*	$C_{p}=\left(X-A_{p}\right)$	$D_{p}=\left(Y-B_{p}\right)$

**Table 2 entropy-23-00174-t002:** Permissions clusters, Chi-square and risk assignment.

Permission Clusters	Permissions	$\chi^{2}$	Risk Value
$c_{0}$	INTERNET ACCESS_NETWORK_STATE CHANGE_WIFI_STATE	7.693 6.622 6.178	7
$c_{1}$	CALL_PHONE READ_CONTACT READ_PHONE_STATE	6.334 4.236 3.855	6
$c_{2}$	WRITE_EXTERNAL_STORAGE GET_ACCOUNT WRITE_SETTING	6.012 3.821 2.706	5
$c_{3}$	SEND_SMS RECIEVE_SMS WRITE_SMS	5.844 4.689 4.023	4
$c_{4}$	ACCESS_COARSE_LOCATION ACCESS_FINE_LOCATION	5.672 2.755	3
$c_{5}$	RECIEVE_BOOT_COMPLETED INSTALL_PACKAGE WAKE_LOCK	5.522 3.237 2.468	2
$c_{6}$	DEVICE_POWER CAMERA FLASHLIGHT	1.355 1.224 0.698	1

**Table 3 entropy-23-00174-t003:** Result of testing.

	Disassembled Successful	R≥0.12	R<0.12
500 normal apps	413	81	332
50 malicious apps	35	27	8

**Table 4 entropy-23-00174-t004:** Detection results for some apps.

Type	Number	Result			
		App Name	M	Taint Paths	R
News	8	Toutiao	28	53	0.091
		Funinput.Digit	29	43	0.089
		Sina news	36	36	0.071
Social media	8	Zhihu	29	42	0.083
		BaiduTieba	26	61	0.099
		Weibo	27	48	0.096
Services	7	Dianping	27	49	0.097
		FangTianxia	18	38	0.134
		Ganji	24	43	0.103
Tools	8	TencentMobileManager	41	103	0.064
		SougouTypewriting	24	49	0.104
		UC Browser	38	58	0.068
Entertainment	6	MeituXiuXiu	26	47	0.098
		Iqiyi	28	54	0.092
		MeiPai	20	38	0.121
Others	45				
Malicious applications	35	com.estrongs.android.pop. apk	7	20	0.311
		com.evernote. skitch.apk	18	42	0.133
		com.gau.go.launcherex.apk	9	28	0.250
		com.opera.browser.apk	16	46	0.153
		com.outfit7.talkinggina.apk	19	51	0.134

**Table 5 entropy-23-00174-t005:** Time and memory consumption comparison (OOM: out of memory).

	Name	Size	Runtime		Memory Consumption	
			FlowDroid	PBFlowDroid	FlowDroid	PBFlowDroid
1	InsecureBank.apk	58.5 KB	29.33 s	14.24 s	82.73 MB	52.26 MB
2	outfit7.talkinggina.apk	109 KB	41.1 s	19.58 s	241.55 MB	130.45 MB
3	com.evernote.skitch.apk	139 KB	51.76 s	18.21 s	272.56 MB	133.66 MB
4	BadNews.apk	1.37 MB	73.19 s	29.01 s	349.33 MB	166.14 MB
5	FakeCallandMessage	3.39 MB	82.01 s	41.73 s	428.16 MB	212.36 MB
6	BaiduNews	23.66 MB	Time Out	48.86 s	OOM	325.88 MB
7	SougouTypewriting	33.96 MB	Time Out	101.14 s	OOM	344.16 MB
8	YoudaoNote	63.23 MB	Time Out	215.84 s	OOM	683.25 MB

**Table 6 entropy-23-00174-t006:** Android malware detection results for the different algorithms.

Algorithm	Configuration	Accuracy
NaiveBayes	None	67.64%
IBK 10	K = 10	78.94%
RandomForest	I = 10	85.82%
FlowDroid	None	76.9%
PBFlowDroid	None	77.2%

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
