# Peer review of "A Modified FlowDroid Based on Chi-Square Test of Permissions"

_entropy, 2021, doi:10.3390/e23020174_

Round 1

Reviewer 1 Report

This article presents a method to cluster Android permissions into permissions predominantly used by malware applications (as opposed to permissions equally popular among benign applications). This knowledge is then used to speed up the analysis process of the static data flow analysis tool FlowDroid by only evaluating flow paths relying on those permissions.

Overall, I recommend to reject this article.

One of the two central contributions of this article is the use of a χ2 test to identify relevant permissions. While this approach seems to be slightly different that others, there is plenty of related work on identifying malware based on their permission use. Hence, this is not new. While the outcome of this article seems to reflect the results of related work, the authors did neither refer to much of nor compare their work to any of the well-known related work in that area (e.g. [R1, R2, R3]). Also, the overall outcome of the permission clustering (table 2) seems to reflect what is already well-known (i.e. that certain permissions are commonly known as risky and often required for malicious purposes, even reflected by risk classification in the AOSP documentation itself).

Moreover the design of the experiment is unclear. What data has been used as the training set (ground truth) to cluster and assess the permission risk value? Was this the same dataset that was later used in the accuracy experiments (section 5.1)? Was there a split between a training set and a validation set? The article suggests that the exact same dataset (consisting of 100 supposedly benign apps from Google Play and 50 known malicious apps from [29]) was used for training (i.e. permission clustering and risk assignment) and also for later validation of detection accuracy. Obviously, an algorithm trained on some data should perform well on exactly that data. Hence, the experiment results only indicate that 77% of apps that are known to be malware can be re-identified as such when only looking at their permissions. The experiment does not validate the potential outcome on apps that have not been used to train the classifier.

The second central contribution of this article is the reduction of analysis complexity of static data flow analysis tool for the tool "FlowDroid". While this seems to be a novel contribution, the presented results are unclear.

  • What is M in table 4?
  • What is the amount of reduction of taint path candidates between original FlowDroid (all paths) and the improved version (likely taint paths)? This would be an important ratio to get context to the results in table 5.
  • What is the meaning of the RS+RC and R values in table 4? Shouldn't there be such values per (taint) path (and not per app)?
  • What are the results for large malicious applications? Do they have more taint paths and therefore make analysis inefficient again?

While the Related Work section lists some potentially related work, there is no comparison between the approaches outlined in this article and any of the identified related work. How are the given methods supperior/different to any of the related work?

Further issues (in order of appearence)

  • (Abstract) Particularly Android devices currently used in the field of automatic control (but also embedded systems with the exception of general-purpose consumer devices, such as smart phones, tablets, TV sets, etc.) seem to have threat models that make them rather resilient against malware since these devices usually come with all there firmware pre-installed and usually do not allow user-installed apps.
  • (Section 2, Related Work) "Arztetet a." should be "Arzt et al."
  • (Section 3.1, ANdroid Permission) "... and the flaws of Android permission mechanism to access or even leak sensitive information." What flaws are you referring to here?
  • (Section 3.2, The FlowDroid) What is Enriched1.apk?
  • (Section 3.3, Mathematical Background) What is A, B, C, and D in formula (1)?
  • (Section 4.2, Risk Calculation) Why are there exactly 7 permissions summed in formula (9)? What are those 7 permissions?
  • For a reader not too familiar with the internas of FlowDroid it is difficult to follow how your approach can perform risk assessment on paths without doing the actual computationally and memory intence part of the analysis work. I believe it would be important to explain those different analysis steps.
  • (Section 5, Experiments) Why are the experiments performed on such outdated hardware and software platforms? E.g. Windows 7 is deprecated and no longer maintained for quite some time now. The latest Java release is Java SE 15 (and the current LTS is JavaSE 11). Also 4GB of RAM seem rather low given current desktop hardware typically has between 8 and 64 GB.
  • (Section 5.1, Accuracy Experiments) What is meant with "shell technology"?
  • Overall, significant English language editing is required.

Missing relevant related work (not exhaustive):

[R1] Sanz B., Santos I., Laorden C., Ugarte-Pedrero X., Bringas P.G., Álvarez G. (2013): "PUMA: Permission Usage to Detect Malware in Android". In: International Joint Conference CISIS’12-ICEUTE´12-SOCO´12 Special Sessions. Advances in Intelligent Systems and Computing, LNCS vol 189, Springer, Berlin, Heidelberg. doi 10.1007/978-3-642-33018-6_30.

[R2] Sun L., Li Z., Yan Q., Srisa-an W., and Pan Y. (2016): "SigPID: significant permission identification for android malware detection". In 2016 11th International Conference on Malicious and Unwanted Software (MALWARE), Fajardo, pp. 1-8, doi 10.1109/MALWARE.2016.7888730.

[R3] Shahriar H., Islam M., and Clincy V. (2017): "Android malware detection using permission analysis". In SoutheastCon 2017, Charlotte, NC, pp. 1-6, doi 10.1109/SECON.2017.7925347.

Reviewer 2 Report

This paper aims to propose a solution that can improve the detection efficiency of FlowDroid based on permissions. The topic looks good and it is practical for FlowDroid. Also, this paper looks OK but needs to be significantly improved before getting it published. Following concerns needs to be addressed:

  1. The presentation is very weak. e.g., "Google Android is a modern operating system for smart device, …", obviously Android is a widely used mobile OS, not only for smart device;
  2. The contributions should be clearly highlighted, you mentioned 'The concept of data at risk and its calculation method are proposed', what does it mean? It is very vague and it will not be a key contributions if you proposed it.
  3. Second contributions, 'An modified FlowDroid (Permission Based FlowDroid, PBFlowDroid) is presented', it is not a contribution as well. Contribution means what is the key point that can demonstrate your work is better than existing works;
  4. Please explain how do you define the 'risk value' in Table 2?
  5. The English presentation needs to be significantly improved.

Reviewer 3 Report

This paper proposed a feature permissions based method to identify the dangerous data flows and improve the detection efficiency of FlowDroid. The experimental results demonstrate the effectiveness of the work. However, the authors should consider some concerns below:
1)In Section V, I don't understand why the threshold is set 0.12 in the experiment. The number of apps is fewer. Typically, malware accounts for about 8%~10% of the total. Hence, the authors should select 500 normal APPs at least.
2)Generally smooth presentation, while containing a number of typos and grammar errors. At least a thorough round of proofreading is indispensable.
3)In Fig.4, many line segments are missing arrows.

Round 2

Reviewer 3 Report

Thanks for improving the manuscript. All concerns are addressed.